# Validating and prioritizing prenatal breastfeeding education recommendations: A nominal group technique study with postnatal mothers and healthcare professionals

Jennifer Kehinde[1], Claire O'Donnell[1], Annmarie Grealish[1,2]*

1 School of Nursing and Midwifery, Health Research Institute, University of Limerick, Limerick, Ireland,
2 Kings Florence Nightingale Faculty of Nursing, Midwifery & Palliative Care, King's College London, London, United Kingdom

* annmarie.grealish@ul.ie

## Abstract

### Introduction

Ireland has the lowest breastfeeding rates in Europe, with only 60% of mothers initiating breastfeeding after giving birth, a figure that drops to 43% three months later, highlighting challenges in promoting the initiation and continuation of breastfeeding due to inadequate antenatal breastfeeding education. Our previous qualitative study revealed a disconnect between idealized prenatal breastfeeding educational approaches and the realities of maternal breastfeeding experiences, prompting a call for practical, interactive education that addresses emotional resilience, societal dynamics, and partner involvement. This study refines and prioritizes these recommendations using the Nominal Group Technique, providing insights for integrating practical content, interactive strategies, and standardized guidance into frameworks aligned with the WHO's 10 Steps to Successful Breastfeeding.

### Methods

This study utilized the Nominal Group Technique to prioritize and evaluate recommendations for improving prenatal breastfeeding education in Ireland, gathering insights from separate sessions with postnatal mothers (n = 6) and healthcare professionals (n = 4). Data analysis was performed using Ritchie's framework for qualitative content analysis and conducted with NVivo (Version 14; QSR, 2024). The study adhered to the Consolidated Criteria for Reporting Qualitative Research (COREQ) to ensure transparency and rigor in its reporting.

### Results

This study identified key recommendations for improving prenatal breastfeeding education through a consensus-building process involving postpartum mothers

**Data availability statement:** The dataset contains sensitive, context-specific information drawn from a small sample, and even with pseudonymization, the risk of deductive disclosure remains. As a result, public access to full transcripts is restricted based on ethical considerations. The anonymized excerpts included in the paper are identified only by general codes (e.g., M1 for postnatal mothers, LCC1 for healthcare professionals) to preserve participant confidentiality. Requests for access to the underlying data may be submitted to the overseeing ethics body, subject to institutional review and appropriate safeguards. The information for the reviewing ethics committee is as follows: Health Service Executive (HSE) Research Ethics Committees of three tertiary hospital sites in the Republic of Ireland (University Maternity Hospital Limerick- REC Ref: 119/2021, Rotunda Maternity Hospital Dublin-REC-2021-027, University Hospital Galway-Ref: C.A. 2718). Data requests can be sent to Dr Annmarie Grealish, Email: annmaire. grealish@ul.ie.

**Funding:** The author(s) received no specific funding for this work.

**Competing interests:** The authors have declared that no competing interests exist.

and healthcare professionals. Mothers emphasized the importance of a balanced approach, including benefits, challenges, and mental support, advocating for personalized learning, interactive group sessions, and standardized guidance. They also highlighted the need for partner involvement and strategies to address societal breastfeeding barriers. Healthcare professionals evaluated the feasibility of integrating these recommendations into Ireland's existing prenatal breastfeeding education framework and their alignment with the WHO's 10 Steps to Successful Breastfeeding. They endorsed the inclusion of balanced educational content and suggested flexible methods, such as recorded sessions, to encourage partner participation and public breastfeeding support. Although there were some differences in implementation strategies, both groups recognized the importance of these recommendations, offering valuable insights for improving prenatal breastfeeding education frameworks.

## Conclusion

This study highlights the critical need to advance prenatal breastfeeding education in a manner that more effectively supports postnatal mothers while maintaining coherence with existing healthcare frameworks. Findings illustrate a shared commitment among postnatal mothers and healthcare professionals to adopt a more balanced educational model that integrates emotional preparedness, personalized learning pathways, and consistent, standardized messaging. By advocating for the inclusion of partner engagement and addressing broader societal challenges, such as breastfeeding in public, the study promotes a comprehensive and contextually responsive approach to improving maternal and infant health outcomes. Grounded in the WHO's 10 Steps to Successful Breastfeeding, the study provides actionable, evidence-based recommendations for enhancing the delivery of prenatal breastfeeding education within the Irish health system.

## Introduction

Promoting breastfeeding has become a global public health priority, with prenatal breastfeeding education recognized as a key strategy for supporting and enhancing breastfeeding practices [1–3]. According to the 2023 assessment by the World Breastfeeding Trends Initiative (WBTI), only an estimated 40% of infants in Ireland continued to receive any form of breastfeeding at three months, while approximately 60% relied exclusively on formula feeding [4]. These statistics highlight a pronounced cultural preference for formula feeding within Irish society. Furthermore, the WBTI, in partnership with Technological University Dublin and UNICEF, released its inaugural report, which ranked Ireland 57th out of 99 participating nations globally, assigning it a score of 56 out of 100 for breastfeeding support and protection [5]. Additionally, Ireland ranks 10th out of 19 European nations evaluated in this study [5]. Furthermore, Ireland consistently reports the lowest breastfeeding initiation rate in Europe, with recent figures indicating initiation at just 60% [5]. This figure is notably lower than

those observed in Australia (90%), the United Kingdom (81%), and the United States (79%) [6–8]. While breastfeeding initiation is high across Europe, exclusive breastfeeding rates decline significantly by six months, with only 25% maintained across the WHO European region with higher continuation in countries like Norway (71%), Sweden (61%), and Germany (57%) [8]. The 2023 Irish Maternity Indicator System report revealed that 64% of babies were breastfed immediately after birth, 61.3% received some form of breastfeeding (exclusive or partial) from birth until hospital discharge, and only 36.9% were exclusively breastfed from birth to discharge [9,10].

In Ireland, prenatal breastfeeding education classes have been established to enhance mothers' knowledge and encourage the initiation and continuation of breastfeeding [11–13]. However, the perceived value and relevance of these classes, as expressed by healthcare professionals and mothers, have yet to be determined, prompting a thoughtful re-evaluation of current prenatal breastfeeding educational approaches [12,14–16]. The prevailing body of evidence within the Irish context has predominantly focused on correlational links between breastfeeding education and outcomes such as initiation, duration, maternal attitudes, and behavioural intent [17–19]. For instance, findings in the study by [17] indicate that breastfeeding outcomes vary by maternal education level, suggesting that future interventions should specifically target mothers with lower education levels and possibly those who are overweight or obese. The study also highlights the potential role of breastfeeding self-efficacy, particularly among older and Irish-born mothers, as a pathway to improved outcomes.

Findings from the study by [18] suggest that positive attitudes, strong breastfeeding confidence, supportive health services, and flexible access to education contribute to successful breastfeeding. However, ongoing social and structural disparities emphasize the need for more equitable breastfeeding support. According to findings in [19], maternal citizenship and ethnicity are among the strongest predictors of breastfeeding initiation and duration. This highlights the possibility that recent improvements in breastfeeding rates in Ireland may be attributed to demographic shifts resulting from immigration rather than solely to the direct impact of national breastfeeding policies or educational efforts [20].

Despite these valuable contributions, existing studies often overlook the perspectives of key stakeholders directly involved in breastfeeding education classes, including mothers and healthcare professionals. Their insights into the structure and content of curricula, the consistency and clarity of messaging, the relevance of covered topics, and the quality of interaction and facilitation remain critically under explored.

The active participation of healthcare consumers in health research is increasingly recognized as a crucial factor in developing impactful and tailored interventions [21–24]. In Ireland, the healthcare system has championed Patient and Public Involvement (PPI) to incorporate diverse perspectives into research and healthcare decision-making [25–28]. This approach is particularly significant in maternal and child health, where decisions about breastfeeding education profoundly influence maternal and infant well-being [29–32]. Involving end-users during the early stages of program development is a well recognized methodological practice, emphasizing the importance of designing interventions that respond to the needs and perspectives of those they aim to serve [29]. Engaging stakeholders early in the process enables future policy adaptations to be more accurately aligned with their realities and expectations, enhancing inclusivity, contextual relevance, and overall impact [30].

This study builds on our previous descriptive qualitative research [33], which identified key areas for potentially improving prenatal breastfeeding education classes in Ireland. Drawing from these insights, the current study aims to validate and enhance the proposed recommendations using the Nominal Group Technique (NGT) [34]. This structured consensus-building method facilitated the prioritization of recommendations and assessed their feasibility for integration into Ireland's existing prenatal breastfeeding education framework. Participants included healthcare professionals (n=4) and postnatal mothers (n=6) who had participated in our previous qualitative study [33], ensuring continuity and depth in stakeholder engagement.

Our previous qualitative study [33] provided insights into the perceptions of postnatal mothers and healthcare professionals regarding prenatal breastfeeding education. The findings revealed that prenatal breastfeeding education is often presented in an idealized manner, which creates a gap between mothers' expectations and their actual experiences.

 

Participants in the study expressed a desire for practical content that balanced the benefits of breastfeeding with realistic discussions about challenges and potential solutions. The study emphasized the importance of a more interactive and personalized educational model within group-based prenatal breastfeeding classes to address the unique needs of expectant mothers. Specifically, mothers in the study [33] suggested using pre-class surveys to identify individual concerns, allowing facilitators to tailor the content and create more relevant and personalized learning experiences, even in group settings.

Furthermore, the limitations of virtual class formats were highlighted in the study [33], with participants noting missed opportunities for engagement and depersonalization. However, they acknowledged the potential of digital tools, such as virtual reality scenarios and progress trackers, to enhance prenatal breastfeeding education. Participants emphasized the importance of integrating interactive elements that foster live discussions, enable shared learning experiences, and provide immediate feedback. The study also underscored the significance of socio-cultural considerations, including increased partner involvement in prenatal breastfeeding education classes and the societal dynamics surrounding breastfeeding in public settings.

Key recommendations from the study [33] emphasized the need to revise prenatal breastfeeding education to reduce unrealistic expectations by incorporating practical content and real-life scenarios. Emotional resilience and mental health support were regarded as crucial for addressing negative breastfeeding experiences, guilt, anxiety and postpartum depression often experienced by mothers due to unmet expectations. Furthermore, enhanced and standardized guidance from well-trained healthcare professionals was suggested to ensure consistency and accuracy in breastfeeding education. Additionally, interventions at the community level that promote partner engagement and a shift in societal perceptions were identified as essential components of a comprehensive strategy to improve breastfeeding outcomes [33].

Building on these findings, this study employed the NGT [34] to engage stakeholders in a structured consensus-building process to gather insights and to determine their alignment with global standards like the WHO's 10 Steps to Successful Breastfeeding [35,36]. NGT is a structured group process that promotes idea generation, prioritization, and consensus-building among participants The NGT is particularly pertinent in this context, where the objective is to capture the diverse perspectives of stakeholders, including postnatal mothers and healthcare professionals, and to achieve a consensus on the most effective strategies for enhancing prenatal breastfeeding education [37,38].

Applying the NGT is particularly advantageous for this study as it bridges the gap between qualitative inquiry and practical implementation [39]. Therefore, the structured nature of the NGT encourages active participation, ensures inclusivity in the decision-making process, and enables the systematic prioritization of ideas, making it an ideal tool for identifying the priorities of postnatal mothers and healthcare professionals in refining future prenatal breastfeeding education.

## Methods

In this study, the recommendations developed in our previous qualitative study [33] were prioritized and assess for feasibility of integration into the current prenatal breastfeeding education framework in Ireland using the NGT [34].Two NGT group sessions were conducted: one with postnatal mothers to prioritize their recommendations and another with healthcare professionals to evaluate the feasibility of integrating these suggestions into Ireland's existing prenatal breastfeeding guide and ensuring they align with the World Health Organization's 10 Steps to Successful Breastfeeding [35].

Qualitative content analysis was performed in both NGT sessions [40]. This method is well-suited to NGT as it enables researchers to systematically categorize and analyse the frequency and significance of participants' responses [41]. This approach enables the identification of themes and prioritization of issues based on group consensus. This aligns well with the NGT's goal of capturing the richness of participant responses and providing an objective overview of the group's prioritised ideas [42,43].

This study is reported in accordance with the Consolidated Criteria for Reporting Qualitative Research (COREQ), utilizing the 32-item checklist to ensure transparency in reporting [44] (S1 Table). Integrating COREQ guidelines alongside the NGT framework highlights the study's commitment to methodological precision and eliciting nuanced, participant-driven insights.

## Study setting and selection of participants

NGT group sessions included participants from three of the four tertiary hospitals (University Maternity Hospital Limerick, University Hospital Galway and Rotunda Maternity Hospital Dublin) in the Republic of Ireland, where participants in our previous qualitative study [33] had been recruited. Eligibility criteria employed purposive snowball sampling methods [45] to selectively recruit participants who had participated in the previous qualitative study. From this pool, six postnatal mothers (n = 6) and four lactation consultants/midwives (n = 4) were purposefully selected to participate in the NGT sessions. The sample size was guided by the methodological requirements of the NGT, which favours small, purposive groups to enable meaningful participation and consensus-building. While not statistically representative, the sample was appropriate for the study's aim of generating stakeholder-informed, prioritized recommendations [46,47].

## Recruitment

Recruitment for NGT group sessions took place between March 2024 and May 2024. The principal investigator (PI) provided the Participant Information Leaflet (PIL) to all potential participants, including postnatal mothers and lactation consultants who had previously participated in our previous qualitative study [33]. The PIL contained detailed information on the study's purpose, features, and the PI's contact information. Establishing direct access to participants was facilitated through their involvement in our previous qualitative study. This prior engagement ensured familiarity and rapport between the researchers and participants, fostering trust critical for meaningful and authentic contributions in the current study. By leveraging this established connection, the study recruited participants with relevant experience and insights, enhancing the depth and quality of the data gathered. Six postnatal mothers (n = 6) and four lactation consultants/midwives (n = 4) participated in the study. Written informed consent was obtained from all participants before data collection. Additionally, verbal consent was reconfirmed immediately before the NGT sessions as an added ethical measure. The respective ethics committees approved this dual consent process.

## Patient and public involvement (PPI)

To ensure that the study remained relevant and responsive to the needs of those it directly impacted, a Patient and Public Involvement (PPI) group was established [48,49]. This group consisted of a deputy director of midwifery, a midwife, and a lactation consultant, all of whom had experience in prenatal breastfeeding education. The PPI group advised on the design and content of the discussion guide for the NGT group sessions (S1 File), using insights and feedback gathered from our previous qualitative study with postnatal mothers and healthcare professionals. This collaborative design ensured that the topics explored during the NGT sessions were firmly rooted in the perspectives, priorities, and concerns expressed by the participants in our previous qualitative study [33].

Through their contributions, the PPI advisors helped shape the study's direction and focus, ensuring that the final prioritized recommendations would be both meaningful and actionable for the community of mothers, healthcare providers, and other stakeholders in prenatal breastfeeding education. We documented the details of the PPI group, including their contributions and impact on the study, in accordance with the Guidance for Reporting Involvement of Patients and the Public (GRIPP2) (S2 Table).

## Data collection

The NGT process began with participants independently generating ideas, which were then shared with the group using a structured round-robin format to ensure equal opportunity for contribution. This was followed by a focused discussion and clarification phase, leading to a final ranking exercise to identify the most preferred options. NGT is widely valued for its ability to encourage creative input, reduce the risk of group-think, and support structured consensus-building [34]. Nonetheless, the method presents limitations, including being time-consuming, potentially restricting open dialogue, and

the risk of facilitator influence. To mitigate these challenges, the principal investigator maintained active engagement with participants, skillfully guided discussions, and fostered an environment conducive to constructive consensus-building.

Data were collected through two online sessions conducted via Microsoft Teams, held approximately three weeks apart. One session was held with postnatal mothers and the other with healthcare professionals, each lasting approximately three and a half hours. The PI expertly guided the discussion sessions, using a discussion guide tailored to each session (S1 File).

Data collection for this study followed the NGT process (Fig 1), designed to refine and prioritize recommendations from our previous qualitative study [31]. The first online session was conducted with postnatal mothers (n = 6), followed by a second session with healthcare professionals (n = 4), each contributing to a structured refinement and evaluation of the recommendations. Before the session, postpartum mother participants (n = 6) received recommendations from our previous qualitative study [33]. Each mother independently assessed and ranked the recommendations, encouraging personal reflection while reducing potential influence from group dynamics. This confidential ranking process was done using a voting card (S2 File), allowing participants to express their priorities without external pressure.

## NGT session with Postnatal mothers (n=6)

### Before the Session

**Stage 1: independent review and ranking.**  Before the session, participants independently reviewed and ranked their recommendations based on their assessments. This preparatory phase allowed participants to critically evaluate and prioritize ideas without external influence, ensuring their contributions reflected personal perspectives (Fig 1).

### During the session

***Stage 2: idea sharing.***  A structured round-robin format was employed during the session to facilitate equitable participation. Each participant systematically presented their prioritized recommendations, ensuring all voices were heard and preventing any individual from dominating. This method fostered a balanced exchange of ideas and encouraged inclusive participation.

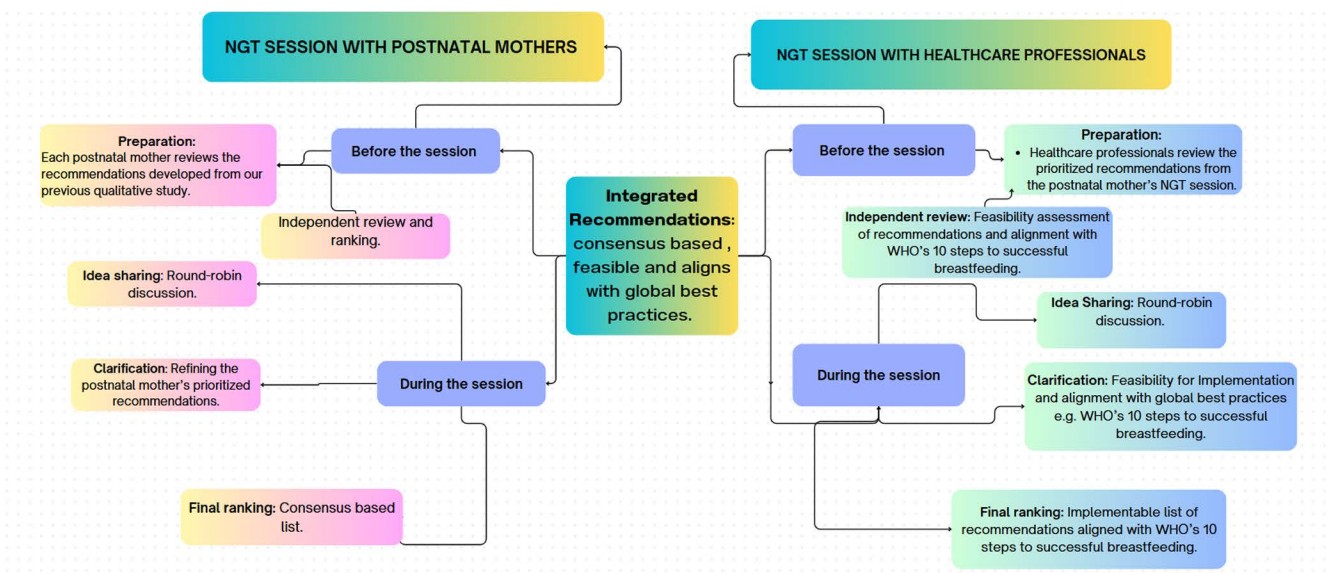

**Fig 1.  NGT Process.**

**Stage 3: clarification.** Following the idea-sharing phase, participants engaged in a clarification stage. This involved collaborative discussions to refine and elaborate on the recommendations, ensuring mutual understanding and resolving any ambiguities. The dialogue improved the clarity and relevance of the proposed recommendations by addressing uncertainties and divergent interpretations.

**Stage 4: final ranking and consensus (generating consensus-based list).** The session concluded with a final ranking exercise, where participants submitted their ranked lists via the chat box. This facilitated a consensus-building process, enabling the group to identify and prioritize the most critical recommendations systematically. The PI aggregated the rankings to create a list of consensus-based prioritization (S3 Table). This ranked list was the foundation for the second session with healthcare professionals.

## NGT session with healthcare professionals (n=4)

A second NGT session was held with four healthcare professionals (n = 4) to evaluate the feasibility of integrating the recommendations prioritized by mothers into Ireland's existing prenatal breastfeeding education framework, and to assess their alignment with the WHO's 10 Steps to Successful Breastfeeding (Table 1).

### Before the session

**Stage 1: independent review and ranking.** Similar to the session with the postnatal mothers, the four participating healthcare professionals were provided with the prioritized recommendations ahead of time to support independent review and ranking. This approach enabled them to evaluate each recommendation based on its relevance and feasibility within their professional context. Rankings were completed privately using a voting card (S3 File). In this preparatory step, the healthcare professionals critically appraised each recommendation's practicality and alignment with the WHO's 10 Steps to Successful Breastfeeding (Table 1).

### During the session

***Stage 2: idea sharing.*** The session began with a structured round-robin format, during which the healthcare professionals systematically presented their evaluations of the recommendations. This approach ensured equitable participation and provided a platform for discussing practical considerations within prenatal breastfeeding education, such as resource availability and constraints.

**Table 1. WHO 10 Steps to Successful Breastfeeding.**

| Steps | Description |
| --- | --- |
| 1a | Comply fully with the International Code of Marketing of Breast-milk Substitutes and relevant World Health Assembly resolutions. |
| 1b | Have a written infant feeding policy routinely communicated to staff and parents. |
| 1c | Establish ongoing monitoring and data-management systems. |
| 2 | Ensure that staff have sufficient knowledge, competence and skills to support breastfeeding. |
| 3 | Discuss the importance and management of breastfeeding with pregnant women and their families. |
| 4 | Facilitate immediate and uninterrupted skin-to-skin contact and support mothers in initiating breastfeeding as soon as possible after birth. |
| 5 | Support mothers in initiating and maintaining breastfeeding and managing common difficulties. |
| 6 | Unless medically indicated, do not provide breastfed newborns any food or fluids other than breast milk. |
| 7 | Enable mothers and their infants to remain together and practice rooming-in 24 hours daily. |
| 8 | Support mothers to recognize and respond to their infants' cues for feeding. |
| 9 | Counsel mothers on the use and risks of feeding bottles, teats and pacifiers. |
| 10 | Coordinate discharge so parents and their infants have timely access to ongoing support and care. |

**Stage 3: clarification.** During this phase, the healthcare professionals engaged in focused discussions to evaluate the feasibility of each recommendation, considering potential resource demands and implementation challenges. Special attention was given to assessing how each recommendation aligned with the WHO's 10 Steps to Successful Breastfeeding. Healthcare professionals collaboratively mapped the recommendations onto the relevant steps, examined their alignment with existing guidelines, and identified areas that may require modification or additional support. This stage facilitated a comprehensive and systematic review of the practical viability of each prioritized recommendation.

**Stage 4: final ranking and consensus (generating consensus-based list).** The session concluded with an open ranking exercise, where participants submitted their ranked lists. The principal investigator aggregated these rankings to produce a consensus-based list that reflected the healthcare professionals' collective agreement on the feasibility of implementing prioritized recommendations and their alignment with the WHO's 10 Steps to Successful Breastfeeding. The final output provided a comprehensive, prioritized roadmap for action (S4 Table).

The PI audio-recorded both NGT sessions. Before each NGT group session, all participants' demographic information was obtained through an online questionnaire via Qualtrics. The Principal Investigator maintained a reflexive journal, fostering transparency, self-awareness, and methodological rigor, which enhanced the study's credibility.

## Data analysis

A modified version of the scoring system developed by [50] was used to prioritize recommendations for integration into Ireland's prenatal breastfeeding education, in alignment with the WHO's 10 Steps to Successful Breastfeeding. Each postnatal mother ranked the recommendations using a descending scale from 5 (highest priority) to 1 (lowest). Aggregate scores were calculated to determine group priorities, with ties resolved by the frequency of votes. Recommendations with cumulative scores between 25 and 30 were classified as high priority (S3 Table).

Subsequently, healthcare professionals assessed these prioritized recommendations based on their feasibility of integration into the current prenatal breastfeeding education framework. Each recommendation was scored on a 5-to-1 descending scale, with higher scores reflecting greater feasibility. Recommendations scoring between 18 and 20 were considered feasible (S4 Table). Additionally, healthcare professionals critically evaluated how each recommendation aligned with the WHO's 10 Steps to Successful Breastfeeding (Table 1). All data from the NGT sessions were transcribed and anonymized to ensure participant confidentiality. The transcripts were then imported into NVIVO (Version 14; QSR, 2024) for data management and coding. Qualitative content analysis [51,52] was employed, combining inductive and deductive coding to capture emerging ideas and align them with predefined study domains that align with the study's objectives and recommendations generated in our previous qualitative study [33]. The principal investigator (PI) conducted multiple readings of the transcripts to ensure a deep understanding of participants' responses and the ideas generated during the sessions. The PI conducted initial coding, followed by collaborative discussions with AG and CO, experienced in qualitative content analysis, to review, refine, and agree upon the emergent themes. Particular emphasis was placed on consensus statements, which served as a basis for identifying key priorities.

Additionally, Ritchie's Framework Analysis [53] explored these prioritized recommendations in greater depth. A matrix (S5 Tables) was developed to chart responses across the two groups (postnatal mothers and healthcare professionals), mapping how participants explained or justified their choices. This approach facilitated the structured summarization of predefined themes while accommodating emergent concepts arising during the sessions. The analysis followed a five-stage process encompassing familiarization, identifying a thematic framework, indexing, charting, and mapping/interpretation. Although the framework analysis followed the five established stages, the charting stage was intentionally repeated (S5 Tables). This iterative step was taken to verify the accuracy and consistency of data categorization across both stakeholder groups. Re-engaging with the charting process allowed for refinement of the matrix entries and ensured that the summaries retained their contextual depth while remaining analytically coherent.

 

### Ethical approval

Ethical approval was obtained from the HSE Research Ethics Committees of three tertiary hospital sites in the Republic of Ireland (Reference Numbers: University Maternity Hospital Limerick- REC Ref: 119/2021, Rotunda Maternity Hospital Dublin-REC-2021–027, University Hospital Galway-Ref: C.A. 2718), and all procedures followed the Declaration of Helsinki regulations [54].

## Results

Table 2 provides demographic details of all six postnatal mothers and four healthcare professionals (lactation consultants/midwives) who participated in the study from three tertiary hospitals.

### Participant characteristics (healthcare professionals and postnatal mothers)

The demographic characteristics of all study participants (n = 10) are presented in Table 2. All four healthcare professionals were female and employed in the maternity units across the selected research sites, where they directly provided prenatal breastfeeding education. All had also participated in the previous qualitative phase of this research. Two participants were dual-qualified as midwives and lactation consultants, while the remaining two identified solely as lactation consultants. Their ages ranged from 36 to 48 years (mean age = 42). Most were of Irish ethnicity (n = 3; 75%), with one identifying as being from another white ethnic background (n = 1; 25%). All were affiliated with either the Nursing and Midwifery Board of Ireland (NMBI) or the Association of Lactation Consultants in Ireland. Educational qualifications ranged from a bachelor's degree (n = 1; 25%) to a master's degree (n = 3; 75%). Their professional experience in healthcare roles spanned 5–6 years, with a mean of 5.5 years.

The postnatal mothers were between 24 and 42 (mean age, 33). The majority of postnatal mothers were cohabiting (n = 4; 66.67%), while the remainder were married (n = 2; 33.33%). In terms of ethnic background, three identified as Irish (50%), one as Black (Irish, African) (16.67%), and two as Asian (Indian) (33.33%). The education status of postnatal mothers ranged from a degree level (n = 1; 16.67%), postgraduate degree (n = 1; 16.67%), second-level education (Leaving Certificate) (n = 2; 33.33%) to Junior Certificate (n = 2; 33.33%). Three postnatal mothers (50%) reported being in full-time employment, two (33.33%) were unemployed, and one (16.67%) chose not to disclose her employment status. Two postnatal mothers (n = 2; 33.33%) identified as Primiparous (first-time mothers), while four mothers (n = 4; 66.67%) identified as Multiparous (Mothers with previous pregnancies).

### Key recommendations and consensus outcomes from the NGT sessions

The NGT sessions systematically evaluated 20 recommendations from our previous qualitative study [33] (S3 Table). These recommendations, which sought to enhance prenatal breastfeeding education classes, were organized around six guiding questions that structured the group discussions and facilitated prioritization: **1)** What do you think would have best prepared you for breastfeeding, considering the need for a balanced approach that addresses both the benefits and challenges, including the mental and emotional aspects like postpartum depression and the frustrations when things don't go as planned? **2)** Given your suggestions—such as using a pre-class survey, offering one-on-one support, implementing a feedback system, and incorporating real-life stories—which interactive strategy would you prioritize to make the classes more engaging? **3)** What digital tools or platforms could enhance the learning experience, considering suggestions like virtual reality for hands-on techniques, discussion breakout rooms, and chat boxes for real-time dialogue? **4)** How can we ensure more consistent and accurate information delivery in prenatal breastfeeding education, given that participants experienced frustration, confusion, and stress due to inconsistent advice? **5)** How important do you think partner participation is in improving the effectiveness of breastfeeding classes? **6)** Would you consider including practical advice on "breastfeeding in public"-specifically within the Irish context —as a priority for future prenatal breastfeeding education sessions? (S3 Table).

**Table 2. Participant demographic information.**

| Study participants | HealthCare Professionals (Lactation Consultants and midwives) | Postnatal Mothers |
|---|---|---|
| **Number of Nominal Group Session Participants** | 4 | 6 |
| **Age Mean (SD)** | 42 (3) | 33 (4.5) |
| **Gender (%)** | | |
| Female | 4 (100%) | 6 (100%) |
| Male | – | – |
| **Years of Experience (%)** | | |
| 2-5 years | 2(50%) | – |
| 6-10 years | 2(50%) | – |
| **Ethnicity (%)** | | |
| Irish | 3 (75%) | 3 (50%) |
| Black | – | 1 (16.67%) |
| Asian (Indian) | – | 2 (33.33%) |
| Other White Background | 1 (25%) | – |
| **Marital Status, n, %** | | |
| Married | – | 2 (33.33%) |
| Co-habiting | – | 4(66.67%) |
| **Educational Attainment (%)** | | |
| Leaving Certificate | – | 2 (33.33%) |
| Junior Certificate | – | 2 (33.33%) |
| Primary School | – | – |
| Degree | 1 (25%) | 1(16.67%) |
| Postgraduate Degree | – | 1(16.67%) |
| MSc | 3(75%) | – |
| **Professional Role (%)** | | |
| Registered Midwife on the Nursing and Midwifery Board of Ireland (NMBI)/ Certified Lactation Consultant on the International Board-Certified Lactation Consultant. | 2 (50%) | – |
| Certified Lactation Consultants on the International Board-Certified Lactation Consultant. | 2 (50%) | – |
| Registered Midwife on the Nursing and Midwifery Board of Ireland (NMBI) | – | – |
| **Employment Status n, %** | | |
| Employed | – | 3 (50%) |
| Stay Home Mother | – | – |
| Unemployed | – | 2 (33.33%) |
| No Response | – | 1 (16.67%) |

*(Continued)*

**Table 2.** (Continued)

| Study participants | HealthCare Professionals (Lactation Consultants and midwives) | Postnatal Mothers |
|---|---|---|
| **Parity (Number of Pregnancies) n, %** | | |
| **Primiparous** (A woman who has given birth once) | – | 2 (33.33%) |
| **Multiparous** (A woman who has given birth to two or more times) | – | 4 (66.67%) |

All participants in this study were either married or cohabiting. No participants identified as single or living alone.

Seven recommendations emerged as the top priorities from the initial 20 recommendations (Table 3). Subsequently, these seven priorities were presented during the NGT session with healthcare professionals to evaluate their feasibility and alignment with the WHO's 10 Steps to Successful Breastfeeding (S4 Table).

### Emergent themes

The findings were analyzed in a structured and rigorous manner to distill core themes and consensus statements from both groups. This involved synthesizing postnatal mothers' educational priorities and comparing them to the feasibility assessments provided by healthcare professionals. This approach entailed carefully triangulating the mothers' consensus priorities with the healthcare professional's feasibility evaluations, particularly noting alignment with the WHO's 10 steps to successful breastfeeding. Areas of divergence and discordance were also examined to reveal subtle differences in perspectives that could guide further refinement of breastfeeding education practices. Therefore, the following sections present the findings and direct quotes to provide a grounded understanding of participant priorities. The S6 Table also provides a summary of the findings.

1. Balanced Prenatal Breastfeeding Education with Emphasis on Mental and Emotional Preparedness (Normalising Frustration and Self-Doubt)

**Consensus Priority from Postnatal Mothers**: Postnatal mothers consistently prioritized a balanced approach to breastfeeding education, emphasizing the need for holistic preparation that includes both the benefits of breastfeeding and its potential challenges; *"If I had known more about the ups and downs, I would not have been so shocked when things got tough" (M1).* Many felt caught off guard by postpartum difficulties, particularly the frustration and doubts that arose when breastfeeding did not go as expected. This realistic framing was seen as essential for setting accurate expectations and offering reassurance.

The mothers emphasized the emotional and mental toll associated with breastfeeding: *"I was not expecting to feel so overwhelmed. If they'd covered that more in the classes, I think I'd have been a bit more prepared" (M3).* They spoke of the *"mental load"* and persistent *"doubts,"* often intensified by the lack of open discussion during classes. Many felt that postpartum support and reassurance could better prepare them for the emotional complexities of breastfeeding. They also expressed the need for prenatal classes to acknowledge and normalize common frustrations and self-doubt: *"It would've been helpful if they'd talked more about how common those doubts are" (M6).* The mothers wanted reassurance that such challenges are typical—and that they are not alone.

**Feasibility of Implementation by Healthcare Professionals** Healthcare professionals agreed that a more balanced approach is feasible to implement, particularly within the WHO Step 3 framework, which covers the benefits and management of breastfeeding. They noted that this addition could be seamlessly integrated, enhancing current messaging; *"Adding a bit about the challenges wouldn't be too hard, right? It's just about tweaking what we're already doing, nothing too drastic"* (LCC1). Other healthcare professionals echoed this sentiment, emphasizing that mothers would benefit from a realistic portrayal that balances encouragement with honesty. This approach combines the benefits of breastfeeding with an in-depth examination of its challenges and complexities.

**Table 3. Prioritized Recommendations from Postnatal Mothers.**

| # | Recommendation | Summary Description | Justification/ Rationale |
|---|---|---|---|
| 1 | Presenting a more balanced approach to breastfeeding education to reflect both the advantages and complexities of breastfeeding, and In-depth discussion regarding the mental and emotional obstacles of breastfeeding, such as "postpartum depression". | Address both benefits and emotional/mental health complexities, including postpartum depression. | Supports informed decision-making and acknowledges real-life challenges experienced by new mothers. |
| 2 | Breakout rooms for discussions and real-time dialogue through indirect communication channels like chat boxes. | Facilitate smaller, safer spaces for mothers to interact and ask questions informally. | Enhances interactivity, participation, and peer support in virtual and online settings. |
| 3 | Group educational sessions that offer opportunities for shared experiences and real-life stories, complementing textbook information. | Move beyond textbook content by incorporating narratives from other mothers. | Increases relatability, emotional validation, and practical understanding of breastfeeding. |
| 4 | Using a pre-class survey to create personalized learning paths. | Assess individual needs in advance to tailor the delivery of education. | Enhances relevance and engagement; supports mothers with varying educational needs. |
| 5 | Establish standardized guidelines for all midwives and lactation consultants and clear communication practices to ensure consistent advice and foster patient confidence by avoiding conflicting messages. | Ensure that all healthcare professionals provide unified, non-conflicting guidance on breastfeeding. | Builds trust in professional advice and reduces confusion or anxiety. |
| 6 | Integrate family-centered education (Partner Involvement) and provide flexible scheduling. | Encourage partners' active role in breastfeeding support by offering options that suit their family routines. | Reinforces the support system and helps sustain breastfeeding efforts at home. |
| 7 | Addressing Breastfeeding in Public: Empowering Mothers Through Practical Support and Open Dialogue | Educate mothers on navigating social stigma and provide real-world tips for public breastfeeding. | Empowers mothers, boosts confidence, and normalizes breastfeeding in diverse public settings. |

While the healthcare professionals recognized the critical importance of emotional support, they discussed the need to gradually incorporate this focus, potentially in Step 5, which supports mothers with breastfeeding initiation and provides ongoing support; *"If we're looking at what's easiest to implement first… the emotional support could come later as part of ongoing care"* (LCC4). This phased approach suggests a feasible strategy for incorporating emotional preparedness into the breastfeeding framework without overwhelming the existing framework for breastfeeding education. Additionally, healthcare professionals strongly supported normalizing discussions on common struggles in the classes, which could help mothers feel more supported; *"We need to normalize those struggles a bit more, so they don't feel like they're on their own"* (LCC3).

**Alignment with Key Stages of WHO's 10 Steps to Successful Breastfeeding** Healthcare professionals agreed that this recommendation aligns most directly with Step 3 of the WHO's 10 Steps to Successful Breastfeeding, which focuses on informing mothers about the benefits and management of breastfeeding. They emphasized that incorporating a balanced and realistic educational approach within this step would enable mothers to understand breastfeeding comprehensively, including its advantages and common emotional and practical challenges. This would help foster realistic expectations, emotional preparedness, and resilience during the postpartum period. In addition, the priority was also seen as closely related to Step 5, which centres on supporting mothers with breastfeeding initiation. Healthcare professionals agreed that integrating structured discussions around emotional and mental wellbeing at this stage would be critical in helping mothers cope with early frustrations, normalizing self-doubt, and reinforcing support during the critical initiation phase. Steps 3 and 5 collectively offer a structured, evidence-informed foundation for integrating this recommendation into Ireland's prenatal breastfeeding education framework.

**Areas of divergence and discordance.** **Divergence** A slight divergence emerged regarding the prioritization of mental and emotional support. While mothers placed this aspect as their highest priority, healthcare professionals recommended a more gradual approach, beginning with educational balance. As one professional noted: *"We don't want to overwhelm them with too much at once, but at the same time, we can't sugar coat it either, we could do it in stages... introduce the benefits early on, and then as we get closer to the due date, bring in more about the challenges and how to handle them, both mentally and emotionally" (LCC2)*. This reflects a difference in immediacy—mothers sought early emotional preparation, while professionals favored a phased integration.

**Discordance** There is no notable discordance, as both groups ultimately agree on the necessity of these topics. However, their views differ regarding the most practical timeline for integration.

## 2) Prioritizing Personalized Learning Through Pre-Class Surveys

**Consensus Priority from Postnatal Mothers**: Postnatal mothers highlighted the value of personalized breastfeeding education and suggested that a pre-class survey could help tailor sessions to their specific concerns: *"It's all about making sure the information" we're getting is actually relevant to what we're dealing with" (M6)*. They viewed this approach as a way to make sessions more meaningful and individually relevant, thereby enhancing engagement and preparedness.

**Feasibility of Implementation by Healthcare Professionals** The healthcare professionals agreed that a pre-class survey would be a feasible first step toward personalization. They recognized it as a manageable addition that wouldn't require extensive restructuring; *"It wouldn't take too much to set up a quick survey before the classes… That way, we could tailor the content a bit more, making it more relevant to them" (LCC1)*. This approach was favored as a practical way of ensuring that the classes are structured to address the core educational needs of the mothers, while also creating personalized learning paths for them, even in a group setting.

**Alignment with Key Stages of WHO's 10 Steps to Successful Breastfeeding** Healthcare professionals identified Step 2 (staff training) and Step 3 (education on breastfeeding benefits and management) as suitable points for integrating the pre-class survey. Under Step 2, staff could be equipped to interpret survey data effectively, while Step 3 would allow these insights to shape personalized educational content. This sequential approach supports targeted, responsive instruction that is aligned with WHO guidelines and better tailored to mothers' needs.

**Areas of divergence or discordance.** No areas of divergence or discordance were identified between postnatal mothers and healthcare professionals. Both groups were aligned in recognizing the value and feasibility of pre-class surveys to improve personalisation and relevance in prenatal breastfeeding education.

## 3) Group Educational Sessions Featuring Real-Life Stories and Shared Experiences

**Consensus Priority from Postnatal Mothers** Postnatal mothers identified real-life stories and shared experiences in group breastfeeding education as a key priority. They found such narratives emotionally validating and practically insightful, helping to normalize the breastfeeding journey: *"When you hear other mums talk honestly about the tough bits and the little wins—it just makes you feel like you're not alone in it" (M5)*. These stories were seen as especially effective in reducing feelings of isolation, fostering peer connection, and offering realistic expectations beyond textbook content.

**Feasibility of Implementation by Healthcare Professionals** Healthcare professionals also recognized the value of authentic maternal narratives in fostering a supportive and relatable learning environment. However, they noted that incorporating structured storytelling would require insight into participants' specific needs*: "If we have a sense of what the mothers are dealing with ahead of time, we can tailor our sessions with real-life and relevant examples" (LCC1)*. Pre-class surveys were viewed as a practical tool to support this personalisation, enabling facilitators to draw on relevant themes or include selected peer stories without overextending resources.

**Alignment with the WHO's 10 Steps to Successful Breastfeeding** This priority was most closely aligned with Step 3, which focuses on informing mothers about the benefits and management of breastfeeding. Incorporating

testimonial-based learning within this step could enrich the educational experience by providing peer-derived insights, enhancing emotional connection, and reinforcing key educational messages. Healthcare professionals noted that these personal accounts could be powerful complements to clinical content, especially when strategically integrated into structured sessions.

**Areas of divergence or discordance.** No significant discordance was observed between postnatal mothers and healthcare professionals regarding the value of shared experiences in breastfeeding education. Both groups supported the inclusion of real-life stories as a means of improving emotional engagement and contextual learning. The only noted divergence is in the approach to delivering these experiences, with mothers emphasizing the natural impact of unstructured sharing, and professionals advocating for guided, thematically linked integration based on pre-identified learning needs from the pre-class survey.

**4) Real-Time Dialogue via Breakout Rooms and Indirect Communication Channels (e.g., Chat Functions).**

**Consensus Priority from Postnatal Mothers** Mothers strongly prioritized using real-time, indirect communication channels, such as chat functions, during breastfeeding education sessions. Rather than viewing these tools as add-ons, they described them as practical enhancements that allow for questions and clarifications to be addressed without interrupting the session or creating cognitive overload. For many, the chat box served as a supportive space where they could quietly pose a question or respond to others without speaking aloud; *"Sometimes you just need to ask something without stopping the whole class—and it's great to scroll back and see what others asked too"* (M4). Mothers also suggested that a dedicated facilitator could monitor the chat to ensure that all responses are acknowledged without overwhelming the flow of verbal instruction.

**Feasibility of Implementation by Healthcare Professionals** Healthcare professionals acknowledged the potential of chat functions to support real-time engagement, particularly for participants less inclined to speak in large groups. However, they ranked this priority lower in feasibility due to concerns about managing dual communication streams during live sessions. They noted that delivering content while monitoring active chat discussions could divide attention and compromise quality: *"It's a great idea in theory, but it's challenging to teach and manage a busy chat simultaneously without missing something important" (LCC2).* Suggested solutions included assigning a secondary facilitator to monitor the chat or using it selectively at designated intervals.

**Alignment with the WHO's 10 Steps to Successful Breastfeeding** This recommendation aligns with Step 3, which emphasizes initiating and maintaining breastfeeding through ongoing support and problem-solving. Indirect communication tools such as chat boxes offer a non-disruptive way to provide immediate clarification, foster engagement, and reinforce breastfeeding guidance during educational sessions.

**Areas of divergence or discordance. Divergence** Both groups recognized the value of real-time, low-pressure communication during group education, but diverged on its feasibility. While mothers viewed chat functions as a manageable, flexible tool to support dialogue without derailing instruction, healthcare professionals highlighted the operational demands of managing this format effectively. Nevertheless, the underlying agreement on the need for responsive, personalized interaction offers an opportunity to explore hybrid models of facilitation, where content delivery is complemented by moderated, asynchronous discussion channels or scheduled Q&A intervals.

**5) Ensuring Consistent and Accurate Breastfeeding Guidance Through Standardization.**

**Consensus Priority from Postnatal Mothers** Postnatal mothers emphasized the importance of receiving clear and consistent breastfeeding guidance from all healthcare providers. Inconsistent or conflicting advice was described as a key source of confusion and anxiety. As reflected in S3 Table, their highest priorities were establishing standardized guidelines and promoting clear, unified communication practices; *"It would help if they had a standard set of guidelines that all midwives and lactation consultants follow." "It's no good getting one bit of advice from one person and then something*

*completely different from another"* (M6). While using a unified resource or handbook was also valued, it was ranked below other strategies, indicating that mothers considered it a complementary tool, rather than the central solution for ensuring consistency.

**Feasibility of Implementation by Healthcare Professionals** Healthcare professionals agreed that consistency across providers was achievable through a multifaceted approach involving standardized training, clear communication strategies, Mentorship, and shared resources. As one explained: *"Standardized training is a must, but we also need to include clear communication strategies in that training. It's not just about what we know, but how we're relaying that information to mothers. We need to be sure we're all explaining things in a way that's clear, consistent, and supportive, no matter who the mother talks to"* (LCC4). While they acknowledged the value of a unified handbook, they emphasized that its effectiveness depended on adequately trained staff who could interpret and apply the content in practice: *"A handbook is a great reference, but without regular training and Mentorship, it's challenging to ensure everyone interprets it consistently. The training really reinforces those guidelines, so were all consistently delivering the same advice to mothers"* (LCC3).

**Alignment with WHO's 10 Steps to Successful Breastfeeding** This recommendation aligns most directly with Step 2, which focuses on staff training to support breastfeeding. The proposed strategies, guidelines, training and communication protocols align with this step's emphasis on ensuring that all staff have the knowledge and skills to provide consistent, evidence-based breastfeeding support.

**Areas of divergence and discordance.** There was no significant discrepancy between the two groups, as both mothers and healthcare professionals agreed on the need for consistent and accurate guidance. The mothers prioritized standardized guidelines and clear communication, and the healthcare professionals acknowledged this, emphasizing that standardized training will aid in achieving consistency. The handbook, though recognized by both groups as useful, was not the central priority for either, but was seen as one component of a broader standardization strategy.

## 6) Partner Involvement as a Supportive Component in Breastfeeding Education

**Consensus Priority from Postnatal Mothers** Postnatal mothers prioritized partner involvement, describing it as a key source of emotional and practical support. They felt that informed partners could help ease breastfeeding challenges by offering encouragement and assistance: *"If the partners are involved, they're not just left on the side lines. They get to understand what we're going through and can offer real support when we're struggling"* (M1).

**Feasibility Concerns from Healthcare Professionals** Healthcare professionals ranked partner involvement lower in priority due to concerns about its feasibility within the constraints of the existing class structure. They cited practical challenges, such as partners' work schedules, which might prevent consistent participation. Instead of in-person involvement, they suggested flexible alternatives, such as recorded sessions, to allow partners to engage with the content at their convenience. Flexible options, like recorded sessions, might work better given the reality of most partners' schedules; *"We want them to be involved, but it has to fit with their availability, so they're not missing out"* (LCC1).

**Alignment with Key Stages of WHO's 10 Steps to Successful Breastfeeding** While healthcare professionals agreed on the potential benefits of partner involvement, they expressed that structured, synchronous integration may not be feasible. Instead, they proposed introducing flexible resources in Step 3, where foundational information about breastfeeding is shared, allowing partners to engage with educational materials at their convenience. This phased approach ideally prepares partners for later stages, such as Step 5 (support during breastfeeding initiation) and Step 7 (continuous involvement through rooming-in), while accommodating logistical constraints.

**Areas of divergence and discordance. Divergence** Differences in perspective emerged regarding approaches to partner involvement. Mothers emphasized the importance of partners being actively and consistently engaged, while healthcare professionals preferred flexible options to accommodate varied schedules. They proposed recorded sessions and online resources as practical alternatives to live, in-person participation, reflecting a contrast with mothers' preference

for synchronous engagement: *"In an ideal world, we would love for partners to join in every session. However, realistically, schedules don't always align, so we must be practical. Offering recorded sessions or online modules would allow them to stay involved without pressure to attend in real-time" (LCC2).*

**Discordance**: Both groups agreed that partner involvement can positively impact breastfeeding support, although they differed in their preferred implementation methods. The variation indicates a difference in practical approaches rather than a core disagreement on the importance of partner support. Both mothers and healthcare professionals recognized that partner engagement is critical in supporting breastfeeding mothers, with the healthcare professionals suggesting flexible resources as a realistic compromise.

### 7) Addressing Breastfeeding in Public: Empowering Mothers Through Practical Support and Open Dialogue.

**Consensus Priority from Postnatal Mothers** Postnatal mothers identified breastfeeding in public as a priority that requires greater emphasis in prenatal education. While they acknowledged that the practice is legally protected in Ireland, they also described emotional discomfort and social pressure as ongoing challenges. Mothers advocated for practical, scenario-based guidance such as role-playing or structured discussions: *"It's grand to be told it's legal, but that doesn't make it any easier when you are out and about" (M4); "I'd love to see more practical advice on how to approach it... to build confidence" (M6).*

**Feasibility of Implementation by Healthcare Professionals** Healthcare professionals acknowledged the cultural sensitivity and relevance of breastfeeding in public, particularly within the Irish context: *"It's definitely more about confidence-building and addressing cultural norms. I'm thinking it might be better not to tie it to any specific step at all" (LCC3).* They agreed on the feasibility of introducing this topic organically through open discussion: *"Instead, maybe we introduce it as a topic for open discussion in the classes" (LCC3).* However, they expressed reservations about formally incorporating it into the WHO 10 Steps to Successful Breastfeeding: *"None of the steps specifically mention public breastfeeding, so it's hard to see where it would fit in without forcing it" (LCC1); "I was thinking the same thing because none of the steps specifically mention public breastfeeding, so it's hard to see where it would fit in without forcing it. Each step has its own focus, and we don't want to dilute their purpose by trying to squeeze this topic in where it might not naturally belong" (LCC2).* Instead, they recommended a more informal approach, allowing the issue to emerge through experience-sharing and class discussions: *"We could bring it up more organically by asking the mothers to share their concerns or experiences about breastfeeding in public" (LCC4).*

**Alignment with WHO's 10 Steps to Successful Breastfeeding**: Unlike the other themes identified in this study, breastfeeding in public does not align directly with any of the WHO's 10 Steps to Successful Breastfeeding. Although Steps 3, 5, and 8 focus on education, support, and responsiveness to infant feeding cues, which were considered potential points of entry, healthcare professionals concluded that the topic does not naturally fit within the framework. As a result, they suggested integrating it through general discussions rather than as a formal curriculum component.

**Areas of divergence and discordance.  Discordance** A clear discordance emerged between postnatal mothers and healthcare professionals regarding the inclusion of breastfeeding in public within the formal prenatal education framework. While both groups acknowledged the significance of the topic, particularly in the Irish context where cultural discomfort and social stigma persist. The mothers advocated for structured, scenario-based preparation during classes to help build confidence and normalize the experience. They viewed this as essential to navigating real-world barriers.

In contrast, healthcare professionals, although supportive of the topic's relevance, expressed concerns about formally embedding it into the existing WHO's Ten Steps to Successful Breastfeeding. They argued that breastfeeding in public does not clearly align with the structure or focus of any specific step, and incorporating it directly could risk diluting the intended content. As a result, they preferred addressing the topic informally through open discussions, rather than through a dedicated curricular component. This discordance reflects a deeper misalignment between maternal expectations and the institutional frameworks guiding education delivery. It underscores the need for culturally sensitive adaptations or

supplementary programming that can address socially embedded barriers without compromising fidelity to international guidelines.

## Discussion

This study expands on the findings of our previous qualitative study [33], which enabled the validation, prioritization, and assessment of the feasibility of specific recommendations through nominal group sessions with postnatal mothers and lactation consultants. These sessions uncovered critical areas for enhancing prenatal breastfeeding education in Ireland, particularly by promoting balanced educational approaches, fostering emotional preparedness, advancing personalized learning, ensuring consistency in guidance, involving partners, and addressing public breastfeeding. The findings align with the WHO 10 Steps to Successful Breastfeeding principles and propose strategies to bridge the gap between established policies and practical application.

A consistent priority for mothers was the need for a balanced approach to breastfeeding education, emphasizing a comprehensive understanding of breastfeeding that incorporates its advantages and potential challenges. Mothers reported that current programs often idealize breastfeeding, leaving them unprepared for the difficulties they might encounter postpartum. Incorporating realistic discussions of breastfeeding challenges aligns well with WHO Step 3, which addresses breastfeeding benefits and management practices. The healthcare professionals agreed that this adjustment would be feasible, requiring only minor modifications to current curricula. Findings from an ethnographic study [55] indicate that women are more likely to feel supported and less marginalized by breastfeeding promotion messages that acknowledge the potential for common breastfeeding challenges. These challenges include difficulties in establishing successful breastfeeding, experiences of emotional distress such as the "baby blues," discomfort associated with breastfeeding in public, and issues related to the premature cessation of breastfeeding [55].

Another critical priority identified was the inclusion of mental and emotional preparedness in prenatal breastfeeding classes. Mothers highlighted the overwhelming emotional toll and self-doubt often associated with breastfeeding, underscoring the need for reassurance and normalization of these experiences. Aligning this focus with WHO Step 5, which supports ongoing maternal care during breastfeeding initiation, presents a practical pathway for integration. While healthcare professionals advocated for phased incorporation of this content to avoid overwhelming participants, the consensus underscores the necessity of addressing emotional resilience as part of a holistic breastfeeding education framework.

A study investigating the relationship between participation in online or in-person antenatal classes and levels of anxiety and depression in Polish women during the COVID-19 pandemic yielded significant findings. It highlighted that maternal anxiety and depression during pregnancy profoundly influence breastfeeding success and child development both prenatally and postnatally. The study emphasized the vital role of antenatal classes in mitigating symptoms of anxiety and depression, fostering well-being in pregnant women, and contributing to improved outcomes for mothers and their children [56].

Both mothers and healthcare professionals emphasized the value of personalizing breastfeeding education through pre-class surveys. The healthcare professionals identified this approach as a manageable addition that could ensure relevance and engagement by tailoring the content to individual needs. They all agreed that the WHO's Steps 2 and 3 provide suitable entry points for implementing this recommendation by training staff to interpret survey findings and adjust educational content accordingly. This strategy aligns with broader educational research, demonstrating personalized interventions' benefits in improving learner engagement and outcomes [57].

Mothers also strongly supported incorporating real-life stories into group sessions, emphasizing their value in providing emotional support and practical insights through shared experience. The mothers believe the narratives will be instrumental in normalizing shared experiences and fostering community among participants. Healthcare professionals acknowledged the potential impact of real-life stories, particularly when paired with pre-class surveys, to ensure relevance. Incorporating these narratives into WHO Step 3, which focuses on educational content, provides an opportunity to create

a more relatable and supportive learning environment. This aligns with existing evidence on the power of storytelling and shared experiences in health education to foster emotional connections and deepen understanding, particularly in creating a supportive and relatable environment for learners [58,59].

An academic journal publication delineating the importance of storytelling as a teaching approach highlights the significant role of storytelling and shared experiences in enhancing learners' absorption and retention of information. The findings demonstrate that this modality fosters a deeper understanding of breastfeeding concepts by engaging participants emotionally and cognitively, creating a relatable and impactful learning environment [60].

The importance of consistent and accurate breastfeeding guidance was another shared priority. Mothers highlighted the confusion caused by inconsistent advice from healthcare providers, advocating for standardized training, clear communication strategies, and shared resources such as handbooks. Healthcare professionals supported these measures, emphasizing the need for training programs to reinforce standardized practices. WHO Step 2, which focuses on training staff, provides an ideal framework for implementing these strategies. Standardized guidance has been shown to improve maternal confidence and foster trust, reducing the potential for conflicting information [61].

Partner involvement emerged as a crucial component of breastfeeding education, with mothers emphasizing the value of informed partners in providing emotional and practical support. While healthcare professionals acknowledged the benefits, they raised concerns about the feasibility of synchronous participation due to logistical constraints. They proposed flexible alternatives, such as recorded sessions and online modules, to accommodate partners' varying schedules. A recent survey investigating fathers' support needs and preferred information sources revealed that mobile applications are among the most favored non-health professional platforms for delivering breastfeeding information [62]. These digital tools provide a practical and accessible means of engaging fathers, empowering them to participate actively in breastfeeding education and support.

Finally, the topic of breastfeeding in public was identified as a critical yet underrepresented area in prenatal breastfeeding education. Mothers sought practical advice and confidence-building strategies to navigate cultural attitudes and personal discomfort. Healthcare professionals acknowledged the significance of this topic but recommended addressing it informally through open discussions, as this recommendation does not seamlessly align with the WHO framework. Addressing the challenges women encounter while breastfeeding in public necessitates coordinated efforts and targeted strategies that consider a comprehensive range of influencing factors. To effectively promote and support breastfeeding, these factors must be acknowledged within a holistic framework encompassing the mother-infant dyad, familial dynamics, the healthcare system, and broader societal and cultural contexts [63]. A cross-sectional study exploring the supportive and challenging aspects of breastfeeding in public among women in Australia, Ireland, and Sweden underscores the need for targeted interventions to address societal barriers. The study's findings reveal that women benefit from comprehensive support in overcoming cultural and social obstacles, enhancing their confidence, and promoting greater acceptance of breastfeeding in public settings [64].

In summary, this study advances the understanding of prenatal breastfeeding education by identifying actionable recommendations that prioritize maternal breastfeeding educational needs and align with evidence-based frameworks. The findings emphasize integrating balanced, personalized, and emotionally supportive approaches into prenatal breastfeeding education. By aligning these recommendations with the WHO 10 Steps to Successful Breastfeeding, healthcare professionals can create a more inclusive and impactful educational framework, bridging the gap between policy and practice.

## Recommendations and implications for practice

This study extends the findings of our prior qualitative study [33], refining and prioritizing recommendations to enhance prenatal breastfeeding education. To mitigate redundancy and ensure progression, the recommendations here are structured as an action-oriented framework for practical implementation, focusing on scalability and adaptability within

healthcare systems. The strategies outlined below aim to translate this research insight into actionable interventions while acknowledging the complexities of integrating these changes into existing practices.

## Developing a holistic and balanced curriculum

A key recommendation emerging from this study is the need to revise existing prenatal breastfeeding education to adopt a more holistic and balanced curricular approach. Current materials often focus predominantly on the biological and nutritional benefits of breastfeeding, inadvertently portraying an idealized version of the experience. Participants highlighted the absence of content that adequately prepares mothers for the psychological and emotional complexities they may encounter during the postpartum period.

To address this, prenatal breastfeeding educational resources should be redeveloped to reflect a nuanced and realistic portrayal of breastfeeding, emphasizing not only its advantages but also its common challenges and complexities. Furthermore, incorporating real-life scenarios and shared experiences can help the mothers set realistic expectations, enhance emotional preparedness, and build psychological resilience. This recommendation aligns with WHO Step 3, which emphasizes equipping mothers with knowledge about the benefits and management of breastfeeding. Consequently, training modules for healthcare educators should be updated to integrate this broader perspective, ensuring coherence and consistency in delivery across clinical and community-based educational settings.

The analytical rationale for this revision is grounded in both this study's findings and existing literature. By addressing the tendency of current programs to idealize breastfeeding, this approach helps prevent the mismatch between expectation and lived experience that often contributes to maternal frustration and diminished confidence in the early postpartum period. A realistic curriculum that balances enthusiasm with authenticity fosters greater maternal self-assurance and enhances preparedness, ultimately supporting a more sustained and satisfying breastfeeding experience [65–67].

## Introducing emotional and mental health preparedness strategies

Another significant recommendation centres on integrating emotional and mental health preparedness within the structure of prenatal breastfeeding education. While existing curricula frequently prioritize physiological techniques and nutritional guidance, they often neglect the psychological demands associated with breastfeeding. This omission leaves many mothers underprepared for the emotional complexities, frustrations, and self-doubt that can arise, particularly during the early postpartum period.

To address this deficit, it is recommended that targeted modules be developed and embedded within prenatal classes, explicitly designed to address the mental and emotional dimensions of breastfeeding. These modules should incorporate facilitated discussions, interactive role-playing exercises, and structured, evidence-based resources to foster open dialogue around emotional struggles. Creating a space in which mothers can articulate concerns and normalize common experiences, such as fatigue, anxiety, or feelings of inadequacy, may mitigate the stigma surrounding emotional distress and provide early strategies for coping.

This initiative can be incorporated into WHO Step 5, which focuses on supporting mothers during breastfeeding initiation. A phased implementation approach is advised, whereby foundational emotional literacy is introduced in the early stages of pregnancy, followed by more comprehensive support strategies as the due date approaches. Such progressive layering ensures that mothers are not overwhelmed while receiving the critical tools necessary for emotional resilience.

This recommendation is grounded in evidence highlighting identified gaps in Ireland's current prenatal education framework. Participants consistently reported a lack of acknowledgement of the emotional realities of breastfeeding in formal education settings. This concern is echoed in the broader literature, which affirms that psychological preparedness is positively associated with breastfeeding success and maternal mental health outcomes [68–70]. Embedding emotional and mental health content into prenatal curricula will ensure that breastfeeding education supports physical techniques and nurtures maternal well-being as a foundational element of postnatal care.

## Enhancing personalization through pre-class surveys

A further recommendation centres on the development and integration of pre-class surveys as a means to personalize prenatal breastfeeding education. This approach responds to the growing recognition that expectant mothers enter educational settings with diverse backgrounds, prior knowledge, and individualized concerns, all influencing their engagement with and absorption of course content. Standardized delivery models may inadvertently overlook these variations, potentially diminishing the relevance and effectiveness of the educational experience.

To address this issue, targeted pre-class surveys are proposed. These instruments would collect key information about participants' breastfeeding knowledge, specific concerns, and preferred learning styles. Facilitators could then use insights gleaned from these surveys to adapt the content, structure, and delivery of sessions to better align with the group's identified needs. This personalized approach would transform the learning environment into one that is more responsive and inclusive. This strategy is directly aligned with WHO Steps 2 and 3, which emphasize the importance of effective staff training and the delivery of comprehensive breastfeeding education. To ensure successful implementation, facilitators would require training in interpreting and applying survey data, equipping them to make real-time instructional adjustments. It is also recommended that pilot testing be conducted in selected clinical or community-based settings to evaluate the survey mechanism's usability, feasibility, and impact.

The analytical rationale underpinning this recommendation is firmly supported by evidence indicating that personalized education enhances participant engagement and optimizes knowledge retention. When learners perceive educational content as directly relevant to their circumstances, they are more likely to internalize key messages and apply them in practice [71,72]. As such, integrating pre-class surveys represents a pragmatic yet impactful strategy to elevate the quality and responsiveness of prenatal breastfeeding education.

## Integrating storytelling and shared experiences

An additional recommendation from this study emphasizes the integration of structured storytelling and shared experiential learning within prenatal breastfeeding education. Mothers participating in the NGT sessions consistently underscored the transformative value of hearing first-hand accounts from others navigating the breastfeeding journey. Such narratives were viewed as emotionally validating and practically instructive, offering nuanced insights often absent from didactic or textbook-based content.

Encouraging participants to share personal breastfeeding experiences in a moderated, group-based setting fosters a culture of mutual support, emotional connection, and peer learning. These storytelling sessions can be further enhanced by utilizing data collected through pre-class surveys, enabling facilitators to identify common themes or concerns and structure discussions accordingly. This alignment ensures that the shared narratives address the specific anxieties and curiosities within the group, thereby maximizing their relevance and impact.

This strategy aligns with WHO Step 3, emphasizing the significance of comprehensive breastfeeding education and management. To effectively implement this recommendation, facilitators should receive training in group moderation techniques that foster psychological safety, inclusivity, and cultural sensitivity. Maintaining a supportive environment ensures participants feel empowered to share personal experiences without fearing judgment or dismissal.

The evidence-informed rationale for this approach is grounded in a substantial body of literature that highlights the pedagogical and Psychosocial benefits of storytelling in creating emotional connections, normalizing challenges, and boosting maternal confidence [59,73]. Additionally, shared experiential learning has been shown to reduce feelings of maternal isolation, which often intensify during the early postpartum period [33,60]. Consequently, integrating storytelling into prenatal education enriches the learning experience and fosters a stronger sense of community and emotional preparedness among participants [71].

## Standardising breastfeeding guidance

This recommendation emphasizes the importance of consistency in breastfeeding education and clinical messaging among all healthcare professionals involved in maternal care. Participants in the study frequently reported discrepancies

in the guidance received from different providers, which led to confusion, reduced confidence, and diminished trust in professional advice. Such inconsistencies, particularly during the early postpartum period, can undermine a mother's sense of efficacy and stability in decision-making around infant feeding. To address this gap, the development of a standardized breastfeeding handbook for healthcare professionals is proposed. This resource should consolidate evidence-based guidance, practical strategies, and communication protocols, serving as a unifying reference for all prenatal and postnatal care staff. To ensure effective uptake, the handbook should be complemented by regular staff training and Mentorship programmes. These sessions would serve not only to reinforce adherence to the standardized content but also to provide a platform for discussing clinical updates and emerging best practices.

This strategy aligns directly with WHO Step 2, which emphasizes the critical role of staff training in delivering high-quality breastfeeding support. In addition, establishing a review committee comprising clinical experts, educators, and service users is recommended. Such a body would be responsible for periodically evaluating and revising the handbook, ensuring that the material remains current, relevant, and responsive to the evolving needs of both providers and service users.

Substantial evidence supports the underlying rationale for this recommendation, indicating that inconsistent or conflicting advice erodes maternal confidence and may deter continued breastfeeding [74,75]. In contrast, standardizing the content and delivery of breastfeeding guidance fosters trust, clarity, and continuity of care, essential for maternal empowerment and positive breastfeeding outcomes [61,76]. Implementing a coherent educational framework thus represents a critical step toward enhancing the overall quality and reliability of breastfeeding support within the healthcare system.

## Encouraging flexible partner involvement

This study's salient recommendation was to integrate partner-focused education within prenatal breastfeeding programmes. Participants emphasized that partners' presence and active support influence maternal experiences and breastfeeding outcomes. Despite this, current educational models often marginalize or overlook partners, failing to provide flexible or relevant avenues for their engagement. To address this oversight, it is recommended that flexible educational resources be developed, tailored to partners' schedules and informational needs. These resources may include recorded sessions, online modules, and on-demand content, allowing partners to access materials at their convenience, regardless of work or care giving obligations. In addition to logistical flexibility, the content must be tailored to address partner support's practical and emotional aspects, highlighting strategies for assisting with infant care, responding to maternal distress, and fostering a supportive home environment.

While participants recognize partner involvement as important, it does not align directly with the WHO's Ten Steps to Successful Breastfeeding. Although aspects of support and education are addressed within the framework, there is currently no explicit provision for structured partner engagement in prenatal breastfeeding education. However, the empirical evidence supporting this recommendation is well-grounded in research linking partner involvement to breastfeeding success, maternal well-being, and the mitigation of postpartum stress [77,78]. Partners with the knowledge and confidence to support breastfeeding contribute to higher initiation and continuation rates. Moreover, flexible and inclusive educational formats help overcome participation barriers, ensuring that partners remain engaged even when constrained by time or distance [79,80]. By positioning partners as integral stakeholders in the breastfeeding journey, this approach reinforces a family-centreed model of care that promotes shared responsibility and sustained maternal support.

## Addressing public breastfeeding challenges

The visibility of breastfeeding in public spaces continues to pose a complex challenge for many mothers, often intersecting with cultural norms, social stigma, and personal discomfort. Despite broader public health efforts to promote breastfeeding, social and environmental barriers persist, inhibiting mothers' ability to feed confidently outside the home. Participants in this study strongly advocated for including targeted, practical guidance within prenatal education to address these challenges explicitly.

Therefore, prenatal breastfeeding programmes should incorporate scenario-based discussions and role-playing exercises to simulate public breastfeeding situations. These strategies enhance mothers' practical preparedness and normalize public breastfeeding as socially acceptable and empowering. Additionally, facilitated group discussions should be embedded within the curriculum to provide a supportive platform for exploring cultural attitudes, personal discomfort, and strategies for self-advocacy. These dialogues allow participants to share concerns, challenge misconceptions, and build solidarity through peer learning. While this recommendation does not align directly with the WHO's 10 Steps to Successful Breastfeeding, it can be effectively integrated into broader educational modules or included as a supplementary component of existing classes. Therefore, facilitators should receive training in cultural competence and inclusive communication to ensure sensitive and effective delivery, equipping them to navigate diverse perspectives and promote respectful dialogue [81].

The empirical basis for this recommendation is well-established. Research has consistently shown that public breastfeeding remains a source of anxiety for many mothers, often contributing to early discontinuation or restricted breastfeeding practices [82,83]. By equipping mothers with practical tools and culturally informed strategies, education can play a transformative role in dismantling stigma and reinforcing breastfeeding as a socially supported norm [80]. This approach enhances maternal confidence and contributes to broader public normalization, helping to shift the sociocultural landscape in favor of breastfeeding-friendly environments.

## Limitations

This study employed the NGT to prioritize stakeholder-informed recommendations effectively. While ideally suited to the study's objective, NGT privileges breadth and consensus over narrative depth, potentially limiting opportunities for extended personal reflection. This could be viewed as a purposeful design feature rather than a methodological weakness. Group dynamics present inherent risks, including dominance by vocal participants or the influence of socially desirable responses. These were mitigated through structured facilitation, anonymous scoring, and independent idea generation. Response bias is also a consideration, as all participants had been involved in the earlier phase of the study. Their sustained engagement may reflect elevated interest or awareness, potentially limiting representativeness. While this enhanced contextual continuity, future studies should seek broader and more diverse sampling to enhance the generalisability of findings and address the inherent limitations of purposive sampling in terms of representativeness.

Finally, given the study's location within the Irish healthcare system, transferability to other cultural or institutional contexts should be cautiously approached. Nonetheless, the themes identified, such as emotional support, consistency in guidance, and partner involvement, will likely have broader relevance and merit further cross-contextual exploration.

## Implications for future research

The findings of this study provide a robust foundation for advancing prenatal breastfeeding education. To ensure effective integration of the proposed recommendations, a targeted feasibility study is warranted to evaluate their practicality, acceptability, and scalability across varied healthcare settings. To support adaptive delivery, pilot implementation should identify resource demands, contextual constraints, and logistical considerations. Evaluating maternal outcomes, including confidence, emotional resilience, and breastfeeding practices, will be essential to assessing the revised curriculum's impact and determining its effectiveness in addressing previously identified gaps.

Stakeholder engagement is critical to this process. Incorporating feedback from mothers, partners, and healthcare professionals will ensure that implementation remains user-centred, contextually relevant, and practically viable. Collaborative input will also support ownership and long-term sustainability. Furthermore, alignment with national policy frameworks will be necessary to embed validated recommendations into breastfeeding education standards. Strategic engagement with policymakers will facilitate system-wide adoption and ensure consistency across care settings. Through rigorous evaluation, stakeholder collaboration, and policy integration, healthcare systems can strengthen prenatal breastfeeding education's inclusivity, personalisation, and impact, ultimately improving outcomes for mothers, families, and communities.

## Supporting information

**S1 Table. COREQ checklist.**
(DOCX)

**S1 File. NGT Discussion Guides.**
(DOCX)

**S2 Table. GRIPP2 short form *(PPI=patient and public involvement).***
(DOCX)

**S2 File. Postnatal Mother's Voting Card.**
(DOCX)

**S3 Table. Postnatal Mothers Priorities Ranking.**
(DOCX)

**S3 File. Healthcare Professionals Voting Card.**
(DOCX)

**S4 Table. Feasibility of Incorporating Prioritized Recommendations into the Prenatal Breastfeeding Education Guide and Their Alignment with the WHO's 10 Steps to Successful Breastfeeding.**
(DOCX)

**S5 Tables. Framework Matrix Based on Ritchie & Spencer's Framework Analysis.**
(DOCX)

**S6 Table. Summary of Findings.**
(DOCX)

## Author contributions

**Conceptualization:** Jennifer Kehinde, Claire O'Donnell, Annmarie Grealish.

**Data curation:** Jennifer Kehinde.

**Formal analysis:** Jennifer Kehinde, Claire O'Donnell, Annmarie Grealish.

**Investigation:** Jennifer Kehinde.

**Methodology:** Jennifer Kehinde, Claire O'Donnell, Annmarie Grealish.

**Project administration:** Jennifer Kehinde.

**Supervision:** Claire O'Donnell, Annmarie Grealish.

**Validation:** Jennifer Kehinde, Claire O'Donnell, Annmarie Grealish.

**Visualization:** Jennifer Kehinde.

**Writing – original draft:** Jennifer Kehinde.

**Writing – review & editing:** Jennifer Kehinde, Claire O'Donnell, Annmarie Grealish.

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
