## [Decision Letter · Decision Letter 0]

PONE-D-25-09271Validating and Prioritizing Prenatal Breastfeeding Education Recommendations: A Nominal Group Technique Study with Postnatal Mothers and Healthcare Professionals.PLOS ONE

Dear Dr. Grealish,

Thank you for submitting your manuscript to PLOS ONE. After careful consideration, we feel that it has merit but does not fully meet PLOS ONE’s publication criteria as it currently stands. Therefore, we invite you to submit a revised version of the manuscript that addresses the points raised during the review process.

We look forward to receiving your revised manuscript.

Kind regards,

Veincent Christian Pepito

Academic Editor

PLOS ONE

2. In the ethics statement in the Methods, you have specified that verbal consent was obtained. Please provide additional details regarding how this consent was documented and witnessed, and state whether this was approved by the IRB

6. We note that this data set consists of interview transcripts. Can you please confirm that all participants gave consent for interview transcript to be published?

If they DID provide consent for these transcripts to be published, please also confirm that the transcripts do not contain any potentially identifying information (or let us know if the participants consented to having their personal details published and made publicly available). We consider the following details to be identifying information:

- Names, nicknames, and initials

- Age more specific than round numbers

- GPS coordinates, physical addresses, IP addresses, email addresses

- Information in small sample sizes (e.g. 40 students from X class in X year at X university)

- Specific dates (e.g. visit dates, interview dates)

- ID numbers

Or, if the participants DID NOT provide consent for these transcripts to be published:

- Provide a de-identified version of the data or excerpts of interview responses

- Provide information regarding how these transcripts can be accessed by researchers who meet the criteria for access to confidential data, including:

a) the grounds for restriction

b) the name of the ethics committee, Institutional Review Board, or third-party organization that is imposing sharing restrictions on the data

c) a non-author, institutional point of contact that is able to field data access queries, in the interest of maintaining long-term data accessibility.

d) Any relevant data set names, URLs, DOIs, etc. that an independent researcher would need in order to request your minimal data set.

For further information on sharing data that contains sensitive participant information, please see: https://journals.plos.org/plosone/s/data-availability#loc-human-research-participant-data-and-other-sensitive-data

If there are ethical, legal, or third-party restrictions upon your dataset, you must provide all of the following details (https://journals.plos.org/plosone/s/data-availability#loc-acceptable-data-access-restrictions):

1. A complete description of the dataset

2. The nature of the restrictions upon the data (ethical, legal, or owned by a third party) and the reasoning behind them

3. The full name of the body imposing the restrictions upon your dataset (ethics committee, institution, data access committee, etc)

4. If the data are owned by a third party, confirmation of whether the authors received any special privileges in accessing the data that other researchers would not have

5. Direct, non-author contact information (preferably email) for the body imposing the restrictions upon the data, to which data access requests can be sent

Additional Editor Comments:

Dear authors, please consider the comments of the reviewers, including my own:

1. The objective of the study is to identify and prioritize recommendations for breastfeeding education. The results should therefore be clearer on what were the recommendations that were identified and how they were prioritized. Now, the results showed emergent themes, which in my opinion, does not answer your objectives.

2. PLOS does not require blinding of previously conducting research and operates via open peer review. There is no need to redact the previous study and might be better if you cite them here so we can assess anything that needs to be assessed.

3. Recommendations should not be bulleted and should still be presented narratively.

4. Please shorten your paper as it is too long as it stands.

5. Add a limitations section to your manuscript.

Reviewers' comments:

Reviewer's Responses to Questions

**Comments to the Author**

1. Is the manuscript technically sound, and do the data support the conclusions?

Reviewer #1: Partly

Reviewer #2: Partly

Reviewer #3: Partly

2. Has the statistical analysis been performed appropriately and rigorously? 

Reviewer #1: Yes

Reviewer #2: No

Reviewer #3: Yes

3. Have the authors made all data underlying the findings in their manuscript fully available?

Reviewer #1: Yes

Reviewer #2: Yes

Reviewer #3: Yes

4. Is the manuscript presented in an intelligible fashion and written in standard English?

Reviewer #1: No

Reviewer #2: No

Reviewer #3: No

5. Review Comments to the Author

Reviewer #1: 1. The sample size of this study is small, and the age, region, economic income, and occupation representativeness of the pregnant woman sample need to be considered. It is advisable to add a description of the rationale for the selection of the sample size, or to clarify the limitations of the small sample size in the discussion.

2. Using Ritchie framework analysis and Nvivo coding, in line with qualitative research specifications, the consensus formation process needs to be supplemented with details, how to solve the parallel ranking.

3. Refine data availability claims (e.g., repository name or access link)

4. Optimize language expressions, correct grammatical errors, and improve readability (e.g. whether "Addin' in a bit about the challenges" should be "Adding a bit about the challenges")

5. If no clear ethical approval document is seen, whether the time of this study meets the ethical approval requirements.

Reviewer #2: This study aims to validate and refine the proposed recommendations from a previous qualitative study to enhance prenatal breastfeeding education classes in Ireland.

Major comments:

1. Interview sample size, selection, and characteristics

The authors report that they selectively recruited participants who had participated in the prior qualitative study. What was the inclusion criteria in the previous study? How many participants were invited of which only 6 postnatal mothers agreed to participate? I’m concerned about the sample size, selection process, as well as the sample characteristics. The authors mention that the low breastfeeding rates underscore a pronounced cultural inclination towards formula feeding within Irish society. Ethnicity may play a strong role in this case and 2 out of 6 postnatal mothers were Asian Indians. Also, in the text they mention Black Irish but in the table Irish and Black are separate categories. I’m wondering about the external validity of the study.

2. Intro: “Irish-based studies on breastfeeding education have primarily focused on the relationship between breastfeeding education and outcomes such as initiation rates, duration, attitudes, and behaviours [16-19].”

What did these studies find? Briefly describe whether education had any significant (positive or negative) on these outcomes.

3. World Health Organization’s 10 Steps to Successful Breastfeeding is integral to the paper and I recommend presenting these steps in a table instead of supplementary file so that readers can relate to the discussion in the paper.

4. Methods section on scoring and ranking is confusing. Postnatal mothers assigned scores but did not rank the recommendations to my understanding. The scores ranged from 1-5 but number of options for each domain ranged from 2-4. There’s a considerable variation in how respondents scored i.e. in S3 table Question 2, participant 1 gave scores (2,1,4,4) whereas participant 2 gave scores (3,2,5,5). Authors need to give more details and clarify this difference in system of scoring.

See scoring vs ranking on page 19 as well: “In the session with postnatal mothers, participants ranked the recommendations using a descending scoring system, assigning 5 points to their highest priority and 1 point to their lowest.”

5. The authors identified 7 recommendations as emerging from the initial list. Table S4 lists these. Why does the summary in S5 have only 6 of these recommendations?

6. The paper needs formatting. The section and sub section headings are not clear. Please number them or clearly use different text size or style to indicate the difference. The numbering of themes in the emergent themes section is inconsistent (1, 1.1, 2, 2.2, 3, 4, 5). Why not number them 1-7?

Themes 1 and 1.1 were part of the same recommendation in the questionnaire as well as tables S4 and S5. Why were they split in the text?

The breakout room theme (part of question 3) is missing from this list altogether whereas this was identified as a priority by mothers.

7. Section 2.1 is “Group Educational Sessions Featuring Real-Life Stories and Shared Experiences” but divergence and discordance discussion is around breakout rooms?

8. Section 3 on Ensuring Consistent and Accurate Breastfeeding Guidance (Pg 26) says “While mothers particularly valued standardized guidelines and a handbook to eliminate conflicting information, healthcare professionals emphasised training and mentorship equally. The healthcare professionals felt that hands-on training and ongoing mentorship would ensure that all staff members effectively interpret and apply the guidelines.”

From table S3, handbook was ranked 3 out of 4 by mothers, why would you say that mothers particularly valued it when they didn’t?

9. On page 27 “The divergence in emphasis reflects a difference in

perspective on the mechanisms for achieving consistency—mothers prioritized a unified resource. At the same time, the healthcare professionals focused on a broader approach, including training

and mentorship.”

Mother did not prioritize a unified source as per table S3. Rather training was ranked second by them and unified source was ranked third after it. Mothers first priority was to establish standardized guidelines and clear consistent communication practices.

On page 30, addressing breastfeeding in public seems more discordant than divergence because the healthcare professions do not believe it fits into the current 10-step guidelines framework whereas the rest of the themes were fitting into the framework but just prioritized differently.

Minor comments:

1. Pg 5: “Participants included healthcare professionals (n=4) and postnatal mothers (n=4)”

Postnatal mothers n=6?

2. Pg 6: “Moreover, community-level interventions promoting partner engagement and shifting societal perceptions were highlighted as essential components of a comprehensive strategy to improve breastfeeding outcomes”

Add period at the end of the sentence.

3. Pg 11: “Data were collected on online on two separate days via MS Teams, approximately 3 weeks apart, with each online session lasting between 3.5 hours”

Please check grammar

4. Pg 16: “Table 1 provides demographic details of all 6 postnatal mothers and 4 healthcare professionals (lactation consultants/ midwives) who participated in the study from three tertiary hospitals (n=10).

Please remove n=10 as it is misleading and redundant.

5. Pg 16, Table 1: Age (mean) should be Age mean (sd)?

6. Pg 18:

“The educational attainment of the healthcare professionals ranged from a Master’s degree (n=3;75%) and a Bachelor’s degree (n=1; 25%).”

“Most of the postnatal mothers were co-habiting (n 4; 66.67%), married (n=2; 33.33%) and identified as Irish (n=3; 50%), Black (Irish, African) (n=1; 16.67%) and Asian (Indian) (n=2; 33.33%).”

“Three of the postnatal mothers identified as in full-time employment (n=3; 50%), unemployed (n=2; 33.33%), and one mother (n=1; 16.67%) did not disclose her employment status.”

Please check grammar for all the sentences listed.

7. Pg 19: “This approach entailed carefully triangulating the mothers’ consensus priorities with the healthcare professional’s feasibility evaluations, particularly noting alignment with the WHO’s ten steps to successful breastfeeding.”

Consider numeric 10 steps instead of ten steps as in the rest of the paper.

Reviewer #3: Overall: this paper is important and although focuses on Ireland's context, I think has global appeal in its findings and the type of analysis. The conclusions from your data are well supported and logical and the recommendations that follow are appropriate given your findings.

Abstract:

If permissible by the journal, in the background of your abstract include the PMID when referencing your earlier study.

Introduction:

- does 40% at three months mean exclusively breastfed or does it include combo feeding? make that clear throughout if you mean exclusive (as in no introduction to formula)

- what is the highest breastfeeding initiation rate in Europe for comparison?

- Checked your numbers from the 2020 report and the stat for 2020 actually says 62.3% not 60%. Please double check your numbers for the other statistics as I don't have access to the journal for the other statistics

-Your intro mentions a need for personalization with breastfeeding education, are these programs you're referring to talking about one on one lactation consultant education, prenatal education from your hospital, education immediately postpartum, group education? Please specify

Recruitment:

- Although rapport and familiarity is nice, I don't know given the nature of your study necessitated it

- how did you determine that 10 participants was enough to collect this type of information? Did you feel like you reached saturation with your data collection?

- also curious the breastfeeding histories of the moms that participated. I imagine there would be differences in opinion for the woman who did extended breastfeeding vs. the woman who didn't continue after leaving hospital

data collection:

- between what and 3.5? was the second session shorter or longer 3.5?

- if participants were going to silently rank and review, why choose a group format and not just a survey with follow up individual interviews (if needed)? What benefit was there to using a focus group vs another method for this type of data collection?

data analysis:

- did you use verbatim transcription for NVIVO?

Table 1:

- In the United States (my location), Black is considered a "race" not an "ethnicity". Does "Irish" mean white here? White would be considered a race. Did your Black participant not come from Ireland? What does any other "white" background mean, like from another European country? Consider a restructure of this variable and break into two variables with 1 for ethnicity if any participants were citizens from another country than Ireland and the other variable to be Race (White, Black, Asian, Another race). Not necessary to note your Asian participant was Indian unless she's from India.

- Marital status- nobody was living solo?

- I can see the benefit of having all the demographics in one table, however I would consider reorganizing the variable order by adding row headers for the mothers and the staff and placing variables under their respective headers

Ideas generated:

-read again to make sure you're not repeating your methods section, omit any repetition

Results:

- i think there is value in including just a very high level summary table of the WHO 10 steps (just what each step is would be sufficient) for reference so someone doesn't have to google it to understand what you are specifically referencing throughout

- It sounds like your healthcare providers are not particularly enthusiastic around public breastfeeding and would only discuss if they needed to based on the quotation selected. Does that feel like an accurate interpretation?

Discussion:

- breastfeeding Peer learning isn't mentioned until the recommendations section but is an evidence based approach to breastfeeding education. Consider expanding on that a little more in the discussion when talking about the importance of storytelling.

Fig 1:

- Your used a step-wise program but this branching figure doesn't apply time particularly well given the arrows are going east/west rather than down. Please restructure for clarity.

Fig 2:

- Does not seem as necessary to me considering it is clearly outlined within the paper itself.

6. PLOS authors have the option to publish the peer review history of their article (what does this mean? ). If published, this will include your full peer review and any attached files.

**Do you want your identity to be public for this peer review?** For information about this choice, including consent withdrawal, please see our Privacy Policy .

Reviewer #1: No

Reviewer #2: No

Reviewer #3: No

---

## [Author Response · Author response to Decision Letter 1]

16 May 2025

See attached file response to Reviewers

---

## [Decision Letter · Decision Letter 1]

PONE-D-25-09271R1Validating and Prioritizing Prenatal Breastfeeding Education Recommendations: A Nominal Group Technique Study with Postnatal Mothers and Healthcare Professionals.PLOS ONE

Dear Dr. Grealish,

Thank you for submitting your manuscript to PLOS ONE. After careful consideration, we feel that it has merit but does not fully meet PLOS ONE’s publication criteria as it currently stands. Therefore, we invite you to submit a revised version of the manuscript that addresses the points raised during the review process. Please submit your revised manuscript by Aug 10 2025 11:59PM. If you will need more time than this to complete your revisions, please reply to this message or contact the journal office at plosone@plos.org . Please include the following items when submitting your revised manuscript:

We look forward to receiving your revised manuscript.

Kind regards,

Veincent Christian Pepito

Academic Editor

PLOS ONE

Journal Requirements:

Reviewers' comments:

Reviewer's Responses to Questions

**Comments to the Author**

1. If the authors have adequately addressed your comments raised in a previous round of review and you feel that this manuscript is now acceptable for publication, you may indicate that here to bypass the “Comments to the Author” section, enter your conflict of interest statement in the “Confidential to Editor” section, and submit your "Accept" recommendation.

Reviewer #1: All comments have been addressed

2. Is the manuscript technically sound, and do the data support the conclusions?

Reviewer #1: Yes

3. Has the statistical analysis been performed appropriately and rigorously? 

Reviewer #1: Yes

4. Have the authors made all data underlying the findings in their manuscript fully available?

Reviewer #1: No

5. Is the manuscript presented in an intelligible fashion and written in standard English?

Reviewer #1: No

6. Review Comments to the Author

Reviewer #1: 1.Sample Representativeness:While the use of a purposive sample is justified for qualitative NGT studies, the authors should further acknowledge the limited generalizability of the findings and discuss this as a study limitation.

2.Language and Structure:The manuscript is written in generally clear English, but some segments—especially those with extensive participant quotations—could benefit from slight condensation or thematic grouping for better clarity and flow.

3.Figures and Tables (Optional):Consider including a visual summary (e.g., a flowchart of the NGT process or a radar chart summarizing consensus priorities) to enhance readability and support international readers’ comprehension.

7. PLOS authors have the option to publish the peer review history of their article (what does this mean? ). If published, this will include your full peer review and any attached files.

**Do you want your identity to be public for this peer review?** For information about this choice, including consent withdrawal, please see our Privacy Policy .

Reviewer #1: No

---

## [Author Response · Author response to Decision Letter 2]

1 Jul 2025

We would like to thank the reviewer(s) for their thoughtful and constructive feedback on our manuscript. We have carefully considered each comment and have revised the manuscript accordingly to improve its clarity, coherence, and scholarly contribution. We believe that the changes made in response to the suggestions have strengthened the paper, and we have provided detailed responses below outlining how each point has been addressed. We appreciate the opportunity to revise our work and hope it will now meet the expectations for publication.

---

## [Editor Report · Decision Letter 2]

Validating and Prioritizing Prenatal Breastfeeding Education Recommendations: A Nominal Group Technique Study with Postnatal Mothers and Healthcare Professionals.

PONE-D-25-09271R2

Dear Dr. Grealish,

We’re pleased to inform you that your manuscript has been judged scientifically suitable for publication and will be formally accepted for publication once it meets all outstanding technical requirements.

Kind regards,

Veincent Christian Pepito

Academic Editor

PLOS ONE
---

## [Editor Report · Acceptance letter]

PONE-D-25-09271R2

PLOS ONE

Dear Dr. Grealish,

I'm pleased to inform you that your manuscript has been deemed suitable for publication in PLOS ONE. Congratulations! Your manuscript is now being handed over to our production team.

Kind regards,

on behalf of

Mr Veincent Christian Pepito

%CORR_ED_EDITOR_ROLE%

PLOS ONE